# Contemporary Evidence and Practice on Right Heart Catheterization in Patients with Acute or Chronic Heart Failure

**DOI:** 10.3390/diagnostics14020136

**Published:** 2024-01-07

**Authors:** Lina Manzi, Luca Sperandeo, Imma Forzano, Domenico Simone Castiello, Domenico Florimonte, Roberta Paolillo, Ciro Santoro, Costantino Mancusi, Luigi Di Serafino, Giovanni Esposito, Giuseppe Gargiulo

**Affiliations:** Department of Advanced Biomedical Sciences, Federico II University of Naples, 80131 Naples, Italy; lina.manzi93@gmail.com (L.M.); luca.sperandeo95@gmail.com (L.S.); imma.forzano@gmail.com (I.F.); ds.castiello@gmail.com (D.S.C.); florimontedomenico@gmail.com (D.F.); robe.paolillo@gmail.com (R.P.); cirohsantoro@gmail.com (C.S.); costantino.mancusi@unina.it (C.M.); luigi.diserafino@unina.it (L.D.S.); espogiov@unina.it (G.E.)

**Keywords:** right heart catheterization, heart failure, HFrEF, HFpEF, acute heart failure, chronic heart failure, advanced heart failure, cardiogenic shock

## Abstract

Heart failure (HF) has a global prevalence of 1–2%, and the incidence around the world is growing. The prevalence increases with age, from around 1% for those aged <55 years to >10% for those aged 70 years or over. Based on studies in hospitalized patients, about 50% of patients have heart failure with reduced ejection fraction (HFrEF), and 50% have heart failure with preserved ejection fraction (HFpEF). HF is associated with high morbidity and mortality, and HF-related hospitalizations are common, costly, and impact both quality of life and prognosis. More than 5–10% of patients deteriorate into advanced HF (AdHF) with worse outcomes, up to cardiogenic shock (CS) condition. Right heart catheterization (RHC) is essential to assess hemodynamics in the diagnosis and care of patients with HF. The aim of this article is to review the evidence on RHC in various clinical scenarios of patients with HF.

## 1. Introduction

Heart failure (HF) has a global prevalence of 1–2%, and the incidence around the world is growing, with more than half of the diagnoses of Heart Failure with Preserved Ejection Fraction (HFpEF). It is also associated with high morbidity and mortality, and HF-related hospitalizations are common, costly, and impact both quality of life and prognosis [1]. Furthermore, more than 5–10% of patients deteriorate into advanced HF (AdHF) with worse outcomes, up to cardiogenic shock (CS) conditions. Right heart catheterization (RHC) is essential to assess hemodynamics in the diagnosis and care of patients with HF. Swan and Ganz introduced a pulmonary artery catheter (PAC) to perform RHC at the bedside, thus opening the era of the use of RHC in patients with HF. Although RHC remains the gold standard for diagnosis and classification of pulmonary hypertension (PH) and to guide its therapeutic management, due to advances in non-invasive imaging techniques in recent years and the contrasting results of trials on the benefits of invasive assessment in HF patients, the use of RHC in these patients has been reduced in clinical practice [2]. However, more recently, the indications for RHC are increasing due to the increasing complexity of HF patients, particularly those presenting with acute HF (AHF), the increasing use of mechanical circulatory support (MCS), the need to guide the timing of advanced therapies, such as left ventricular assist device (LVAD) implantation or heart transplantation in patients with HF with reduced ejection fraction (HFrEF), and the introduction of new diagnostic algorithms for HFpEF patients [3,4]. Indeed, RHC monitoring of hemodynamics has been incorporated into CS treatment algorithms or HFpEF diagnostic algorithms. Furthermore, an invasive hemodynamic assessment is essential to LVAD implantation or heart transplantation candidacy in patients with advanced HF.

Although RHC remains a historical technique in interventional cardiology, its role during the last decades has changed with a trend to limit its use in daily practice. However, when properly performed, it can offer important diagnostic and prognostic information, and there has been growing literature in the last few years supporting its use in the setting of HF patients. Therefore, the aim of this article is to review the current understanding and indications of RHC in various clinical scenarios of patients with HF.

## 2. Right Heart Catheterization: Principles and Methodology

RHC is an invasive hemodynamic procedure that allows for the direct measurement of the right-side cardiac chamber-filling pressure and assessment of the cardiac output (CO).

In 1929, aiming to find better approaches for delivering drugs directly into the heart, Werner Forssman performed the first RHC on himself by advancing a well-oiled 4 French ureteral catheter via the left cubital vein for a total length of 65 cm and then walking the stairs to the radiology department to perform a chest X-ray and document the right atrial catheter position [5]. Subsequently, the American physician-physiologists André Cournand and Dickinson Richards redesigned Forssmann’s catheter and advanced the technique in the 1940s, allowing for a safer procedure, longer indwell times, and easy, repeated collection of true mixed venous blood and, consequently, the calculation of cardiac output by the use of the direct Fick principle for the first time in humans. The three physicians were awarded the Nobel Prize in Physiology and Medicine for these contributions in 1956. 

However, it was not until 1970 that Jeremy Swan and William Ganz developed their eponymous PAC while studying and revolutionizing the measurement of CO, pressures within the left side of the heart, and resistance in systemic and pulmonary circulations. They added a balloon to the catheter tip of a standard pulmonary catheter, which allowed bedside placement via floatation and also provided an opportunity to measure pressure in the right atrium (RA) and the pulmonary arteries (PA) continuously and developed the idea of the thermistor at the tip, which allowed for the direct measurement of CO using the thermodilution technique. Because of this catheter’s widespread use thereafter, the catheter became commonly known as the “Swan–Ganz” catheter [6,7,8].

The use of the RHC has decreased in the last 30 years because of recent advances in less invasive hemodynamic monitoring (IHM) techniques, combined with the publication of the results of several randomized trials that failed to show improvements in outcomes with the use of the RHC in various settings [9]. Today, RHC remains the best technique for PH diagnosis and, in recent times, has become the gold standard test to establish the diagnosis of HFpEF [1].

RHC is ideally performed with minimal conscious sedation in supine patients in euvolemic status [10,11]. Once the patient is on the table and the sterile field is established, venous access should be obtained. Internal jugular, brachial, or femoral venous access can be utilized. Venous access can be safely obtained using the Seldinger technique. An 18- or 21-gauge micropuncture needle is used to cannulate the venous under ultrasound guidance, followed by a 0.018″ 40 cm fine wire. Once the venous placement is confirmed, a 5 Fr catheter is inserted, through which a 0.035″ wire can be advanced and after an anatomical evaluation of a correct position, a special, dedicated 5 or 7–8 Fr sheath with a valve is inserted into the lumen, enabling easy access into the vein [12,13]. The Swan–Ganz catheter is a balloon-tipped quadruple-lumen catheter with a thermodilution sensor attached to a pressure transducer outside the body. This transducer allows for the measurement of right-sided cardiac pressures. The Swan–Ganz catheter is composed of four proximal ports, a body, and a tip with the balloon. The proximal lumen, blue lumen, or central venous pressure (CVP) port represents the right atrial lumen. It is the proximal port and can be used for infusion. This port can assess CVP and right atrial pressure (RAP). The yellow lumen or PA distal is the pulmonary artery lumen; it is the distal port at the tip of the catheter. This port is connected to a transducer and allows the measurement of the pulmonary artery pressure (PAP). Mixed venous can also be drawn from this port. The thermistor is a red/white connector that consists of a temperature-sensitive wire that terminates 4 cm proximal to the tip of the catheter. The terminal portion of the wire is called a thermistor bead, and it rests in a main pulmonary artery when the catheter tip is positioned correctly. The connection of the thermistor port to the CO monitor allows the determination of the CO using thermodilution. At the distal tip, the S-G catheter has a balloon that can be inflated and helps the clinician place the tip of the catheter in the PA. The balloon, when inflated, causes the catheter to wedge in a small pulmonary blood vessel. So wedged, the catheter can indirectly measure the pressure in the heart’s left atrium (LA).

The Swan–Ganz catheter is the catheter of choice for the jugular approach, whereas a so-called pulmonary hypertension Swan catheter is advisable for the femoral approach and for patients with PH or severe tricuspid regurgitation (TR) because it is stiffer and more torquable [14].

The standard Swan–Ganz catheter is advanced through the sheath into the vein using the internal jugular vein as access. The balloon is inflated after the catheter is advanced to roughly 15 cm so to avoid inflating it within the access sheath. Balloon inflation will then make advancing the catheter to the RA much easier with reduced perforation risk. The catheter is advanced under fluoroscopic guidance without a wire unless there is difficulty with manipulation [6,8,14]. When the catheter reaches the RA, a pulsatile right atrial waveform is observed. Prior to recording pressures, a reference is established by zeroing the system. Zeroing involves opening the air-fluid transducer to air to equilibrate with atmospheric pressure. When this is being performed, the air–fluid transducer must be at the level of the heart. To establish uniformity of the pressure transducer setting, the European Society of Cardiology (ESC) and the European Respiratory Society (ERS) recommend performing the zeroing for the pressure transducer at the mid-thoracic line (with a suggested reference point defined by the intersection of the frontal plane at the mid-thoracic level, the transverse plane at the level of fourth anterior intercostal space and the midsagittal plane) with the patient in a supine position, halfway between the anterior sternum and bed surface, which represents the level of the LA [15]. After zeroing, once the RAP waveform is obtained, the catheter is manipulated to turn towards the right ventricle (RV), and right ventricular pressure (RVP) is obtained. Following this, the catheter is usually advanced to a wedge position in PA to measure the pulmonary capillary wedge (or occlusive) pressure (PCWP). The tip of the catheter lies in the main pulmonary artery, where the balloon can be inflated and deflated for the measurement. The balloon can be inflated here to obtain PCWP, which gives an indirect assessment of the filling pressures of the left side. Once this is done, the balloon can be deflated and brought back a few centimeters into the PA, where the PAP can be recorded. The PA blood sample is withdrawn using the distal yellow port, and mixed venous oxygen saturation is obtained. Arterial saturation must be obtained separately so as to determine the CO using the Fick method. Thermodilution can be performed by injecting cold saline into the proximal blue port into the RA, where it mixes with blood, and the temperature difference is detected by a thermistor. Thermodilution should be repeated a minimum of three times to obtain an average CO and cardiac index (CI). The PAC can be manipulated and placed in the superior or inferior vena cava for withdrawing blood samples and estimating oxygen saturations [11].

Hemodynamic parameters assessment should be recorded at the passive end-expiration phase and should be averaged over three cardiac cycles in patients with sinus rhythm (more cycles in those with atrial fibrillation). The RHC enables assessment of the following variables: mean RAP, systolic and end-diastolic RV pressure, systolic, diastolic, and mean pulmonary artery pressure (mPAP), PCWP, mean arterial pressure (MAP), mixed venous oxygen saturation (SvO2), and CO (Figure 1; Table 1).

The first chamber crossed is the right atrium. The right atrial pressure curve has five deflections—three positive and two negatives. An a-wave (positive) is a diastolic wave due to cardiac contraction of the atrium. Therefore, it is absent in the case of atrial fibrillation. A c-wave (positive) is due to the protrusion in the atrium of the closed atrioventricular valve during ventricle isometric contraction; it is generally not visible. An x-wave (negative) is derived from the pressure drop in the atrial cavity caused by ventricular systole. A v-wave (positive) is produced by the increase in atrium pressure secondary to the filling of the atrium when the atrioventricular valve is still closed during ventricular systole. It is typically very high and visible in the case of significant MR. The y-wave (negative) derives from the pressure drop caused by the opening of the atrioventricular valve, which allows atrial emptying and rapid ventricular filling.

RVP is identified by a pulsatile pressure curve with a systolic and diastolic phase. The subsequent entry into the pulmonary artery is characterized, on the pressure curve, by a significantly increased diastolic pressure, an expression of the closure of the pulmonary valve in the absence of substantial changes in the systolic peak. When the tip of the Swan–Ganz catheter is wedged, the balloon occludes the distal pulmonary artery, completely obstructing blood flow, so a static column of blood will form between the tip of the Swan–Ganz catheter and the atrium, whereby the pressure wave detected at the tip of the catheter will be equivalent to the left atrial pressure. 

The PCWP is a typical non-pulsatile atrial wave. When measuring PCWP, it is advised that the balloon should be inflated in the right atrium and then advanced until reaching the PCWP position [16]. Importantly, repeated inflations and deflations in the wedge position are not recommended to avoid the risk of pulmonary artery rupture [16]. PCWP should be subsequently recorded as the mean of three measurements at end-expiration and avoiding breath-holding or the Valsalva maneuver [16]. The latter recommendation arises from the observation that PCWP should be evaluated at a functional residual volume when the intra- and extra-thoracic pressures are equal. PCWP may be significantly influenced by respiratory swings, which affect intra-thoracic pressure. Since these variations are marginal at the end of normal expiration, the measurement of end expiratory PCWP minimizes this effect [16]. However, there are cases in which this recommendation may not be applicable, such as patients with chronic obstructive pulmonary disease (COPD), in whom there is often a prominent swing in intrathoracic pressure affecting intracardiac pressures and in whom some data suggest that more reliable PCWP pressures might derive from the average over the entire respiratory cycle [17].

To verify that the catheter position is appropriate and PCWP value is reliable, some practical aspects should be considered: (a) curve morphology changes from pulsatile to a typical non-pulsatile atrial curve in which the a- and v-waves are recognized; (b) PCWP curve has variations related to respiratory activity with a clear respiratory swing that should be visible; (c) oxygen saturation test: a truly wedged catheter will yield a mixed venous oxyhemoglobin saturation reflective of the postcapillary pulmonary bed, typically greater than 90% to 95%, while lower values should prompt repeat attempts to wedge, including alternate vascular areas or consideration of direct left ventricle (LV) measurement; (d) mean PCWP should be equal to or lower than diastolic pulmonary arterial pressure or however should not exceed by more than 1 to 2 mmHg diastolic pulmonary pressure unless the patient has significant pulmonary congestion or moderate to severe MR; the presence of large v-waves (from mitral regurgitation or left heart disease) or atrial fibrillation often leads to a mean PCWP that is greater than the end-diastolic PCWP, and in such cases, the a-wave of the PCWP tracing should be measured; (e) the catheter tip position must be stable on fluoroscopy; (f) hold-up of contrast in the distal circulation should occur, however, forceful aspiration or injection of contrast or saline through the end-hole in the occlusion position should be avoided to prevent complications [17].

Aside from pulmonary pressures during RHC, CO may be calculated through the Fick method and or thermodilution method. The Fick method is based on the principle of conservation of mass; the quantity of oxygen absorbed by the lung, therefore consumed by the subject, should be equal to the arteriovenous difference in oxygen. However, direct measurement of oxygen consumption (direct Fick method) takes time, and suitable instrumentation is not available in most laboratories. Therefore, more frequently, an indirect Fick method where oxygen consumption is estimated with mathematical formulas that consider body surface area, age, and sex.

For this calculation, SvO2 is measured while the tip of the catheter is in the RA or PA, whereas systemic oxygen saturation is often acquired noninvasively by oximetry. However, this method is less reliable compared to the other techniques and is associated with underestimation or overestimation of CO of about 10–15%; therefore, it is not the preferred method for the measurement of CO. The thermodilution method consists of injecting 10 mL of cold physiological solution (0–4 °C) into the RA through the proximal port of the Swan–Ganz catheter and measuring the temperature change in the PA by means of a thermistor placed on the distal part of the catheter. The transient decrease in PA temperature will be described by a temperature-time curve, and the area under the curve will be inversely proportional to CO. The injection must be done quickly, steadily, and progressively. A reliable measurement corresponds to the average of three consecutive measurements spaced out for at least 90 s to allow the basal temperature to be restored in the right heart chambers. The advantage of the thermodilution method is the relative ease of use and results. Still, it is less accurate in patients with significant tricuspid or pulmonic regurgitation, intracardiac shunts, low cardiac output, or irregular rhythms.

In clinical practice, the assessment of the CO using the thermodilution method is preferred to the evaluation by the Fick method. The latter is more accurate in the case of TR and low cardiac output, while it should not be used in patients with significant mitral or aortic regurgitation and is unsuitable under conditions of rapid changes in flow, and patients should not be on supplemental oxygen. The main limitation of the Fick method is caused by the fact that the bedside measurement of oxygen uptake is technically demanding [16].

The SvO2 can be used to quantify oxygen consumption, and his assessment is obtained either intermittently by drawing blood samples or continuously using a fiberoptic catheter. Furthermore, CI is calculated by dividing CO by body surface area (BSA). Pulmonary vascular resistance (PVR) is calculated by dividing the transpulmonary gradient (TPG) by CO. The TPG is calculated by subtracting the mean pulmonary artery pressure (mPAP) from PCWP.

Hence, the RHC provides information on blood flow and its adequacy (CO and SvO2), filling pressures of left and right ventricles and right ventricular after load. It is one of the most integrative tools for evaluating cardiovascular function [9,16].

At last, the complications of RHC are rare, most of which are related to central venous access (<3.6%) and depend on operator experience. More serious arrhythmias, heart block (0.3% to 3.8%), pulmonary artery rupture (0.02% to 0.03%), and right ventricular free wall rupture (<0.01%) are very rare [18].

## 3. Right Heart Catheterization in HFpEF

HFpEF is a syndrome characterized by multiorgan systemic involvement in the presence of a LVEF > 50%. It is increasing in prevalence as it is related to several comorbidities such as advancing age, hypertension, diabetes mellitus and obesity. HFpEF is related to increased morbidity and mortality. Patients with HFpEF have a higher mortality rate and worse Quality of Life (QoL) compared with patients with HFrEF [19]. Considering the high and increasing incidence of HFpEF, combined with an objective difficulty of the diagnosis, scores have been proposed to help clinicians in the diagnostic process. Two scores have been recognized as of great utility and interest: the HFA–PEFF [4], a diagnostic algorithm proposed by the Heart Failure Association (HFA) of the ESC and the H2FPEF proposed by Reddy and colleagues [3]. The latter was validated using invasive exercise hemodynamic parameters, and it is easier for clinicians to use. H2FPEF score uses six parameters that are quickly accessible: weight (body mass index [BMI] > 30 kg/m^2^), hypertension, atrial fibrillation, PH (estimated sPAP > 35 mm Hg on Doppler echocardiography), age > 60 years, filling pressures (E/e′ > 9 on Doppler echocardiography). A score of 6 or more is suggestive of HFpEF [3]. The pros of this score are that it can be easily assessed, even at the patient’s bedside, and it is reliable to rule in or rule out the diagnosis of HFpEF. However, H2FPEF does not include natriuretic peptide (NP) values, which may be problematic, as NPs are a component of the Universal Definition of HFpEF [20] (Table 2). Instead, the HFA of the ESC [4] has proposed a step-by-step diagnostic approach, the “HFA-PEFF diagnostic algorithm,” from initial clinical assessment to more specialized tests to evaluate patients with HFpEF (Figure 2). The first step includes a detailed clinical and familiar history, an electrocardiogram (ECG), blood tests, standard echocardiography and exercise tests. In detail, several tests are recommended: sodium, potassium, kidney and liver function tests, HbA1c, thyroid stimulating hormone, and full blood count to diagnose anemia, which may aggravate symptoms in HFpEF patients [21,22]. NP may help distinguish cardiogenic dyspnoea. However, normal NP levels are frequent in HFpEF patients, especially in the presence of obesity [23,24]. Standard echocardiography is important to exclude alternative causes such as coronary artery disease, valvular disease, pericardium, and pulmonary disease [25,26]. Non-dilatated and concentrating hypertrophic LV with a normal EF and left atrial enlargement are typical features of HFpEF. Therefore, for the final score, it is essential to calculate more detailed parameters such as average septal–lateral E/e′ ratio, TR peak velocity and pulmonary arterial systolic pressure, left ventricular global longitudinal systolic strain, left atrial volume index, left ventricular mass index, and relative wall thickness [4]. However, echocardiographic assessment has been shown to have a less predictive capacity to evaluate intracardiac pressure than invasive assessment. In particular, the correlation between echocardiographic predictor of LV filling pressures, E/e′, and PCWP or Left Ventricular End Diastolic Pressure (LVEDP), invasively evaluated, has been demonstrated to be poor [27]. 

Furthermore, it is important to perform an exercise test on these patients to assess epicardial stenotic coronary artery disease (CAD) that may coexist in patients with HFpEF, may be part of HFpEF pathophysiology [28], and may impact mortality [29]. A stress test is indicated if CAD is suspected, such as a bicycle or treadmill exercise test, dobutamine stress echocardiography, cardiac magnetic resonance (CMR) imaging, or myocardial scintigraphy [30]. Furthermore, a stress test provides information about exercise capacity, blood pressure, heart rate response, and chronotropic incompetence, which are present in 33–77% of HFpEF patients [30]. The score includes morphological, functional, and biomarker parameters (Figure 2). A score of >5 points is diagnostic of HFpEF, while a score of <1 point is very unlikely the diagnosis of HFpEF. Patients with an intermediate score need further evaluation [4]. RHC remains the gold standard test to establish the diagnosis of HFpEF [4,31,32]. An invasively measured PCWP of ≥15 or LVEDP ≥ 16 mmHg is generally considered diagnostic of HFpEF [1] while concurrently ruling out PH group 1. According to recent guidelines, the patient with PH group 1 has mPAP > 20 mmHg, PCWP ≤ 15 mmHg, PVR > 2 Wood units [33]. Additionally, left heart catheterization (LHC) may be useful to assess left ventricular pressures (LVEDP and LVESP) and, together, elastance, using pressure-volume (PV) loops. In patients with HFpEF, higher LVEDP is found during LHC (at rest, but most of all during exercise), and usually, the PV loop shows an upward and leftward shift [34]. However, a time constant of LV relaxation (tau > 48 ms) [35], measured by high-fidelity pressure catheters, is a demonstration of impaired LV relaxation. This, along with LV filling pressures at rest (LVEDP > 16 mmHg), confirms definite evidence of HFpEF [4]. 

Furthermore, LHC could be useful to assess or exclude concomitant CAD or other conditions that could mimic HFpEF (i.e., constrictive pericarditis). Of course, in these cases, an invasive coronary angiography is necessary for definitive diagnosis because it allows for detecting obstructive epicardial CAD as well as evaluating coronary microvascular dysfunction (CMD) by measuring indexes like coronary flow reserve (CFR) and index of microvascular resistance (IMR) in relatively easily way by using a coronary wire with temperature and pressure sensors and injecting saline flush at ambient temperature to measure these indexes by the thermodilution method. The evaluation of CMD could play a relevant role because it has been hypothesized to be one of the mechanisms underlying HFpEF. Therefore, since epicardial CAD affects myocardial perfusion, it is challenging to clarify the relationship between HFpEF and MVD in cohorts with highly prevalent epicardial CAD. Invasive studies show that patients with HFpEF have a high prevalence of CMD, ranging between 70% and 85% depending on the diagnostic thresholds used, which vary between studies: CFR ≤ 2 to ≤2.5, IMR ≥ 25 [36,37].

However, a large proportion of HFpEF patients have normal non-invasive and invasive tests at rest. Indeed, in most patients with HFpEF, due to impaired early diastolic relaxation and poor LV compliance, volume changes lead to larger increases in LVEDP during exercise. Additionally, the exercise-induced increase in cardiac output is reduced due to poor contractile reserve and chronotropic incompetence [10,38,39]. In these cases, exercise RHC is the gold-standard diagnostic test for HFpEF [40]. Protocols differ slightly between sites but generally include increased supine or standing workload, measurements of RAP, PAP, and PCWP, and the measurement of CO using either direct Fick or thermodilution [41,42]. The protocol starts with an unloaded cycle (0 watts) and increases by 20–25 Watts up to maximal exercise capacity [4] (Figure 3). Stages should be held for 2–3 min to allow for hemodynamic stabilization. It is impossible to record pressures at the end-expiration during the exercise; hence, they should be measured after at least three cardiac cycles. If CO is measured by the thermodilution method, two measurements should be performed so the total duration per stage becomes 5–7 min [43].

Furthermore, after discontinuation of the workload, parameters should be measured during the recovery period. In some centers, these measurements are performed through a jugular vein catheter in the context of a cardiopulmonary exercise test (CPET) (invasive CPET). Patients with peak exercise PCWP ≥ 25 mmHg are classified as having HFpEF [4]. To describe the PCWP response to exercise, the ratio of PCWP at peak exercise to workload normalized to body weight can be calculated [PCWL (mmHg/W/kg)] [4]. This variable is less dependent on the patient’s cooperation than PCWP at peak exercise because, in patients who stop exercising prematurely, peak PCWP may be misleadingly low, while PCWL can still detect abnormal hemodynamics.

Patients with “high” exercise PCWP have poor outcomes with increased mortality [42,44,45]. However, it has been demonstrated that healthy and elderly individuals may exceed “normal” LV filling pressure parameters during exercise [46,47]. Hence, to overcome these limitations, a novel method called the PCWP/CO slope has been developed and validated to evaluate LV performance during exercise RHC [48,49]. This diagnostic value has only been validated with the Fick method. An exercise PCWP/CO slope > 2 mmHg/L/min has more diagnostic sensitivity and specificity compared with the peak PCWP criteria for diagnosis of HFpEF and better predicts the risk, as it considers the whole workload spectrum [49]. Furthermore, an exercise PCWP/CO slope > 2 mmHg/L/min has been associated with poor functional capacity and adverse clinical outcomes in HFpEF patients [49,50]. However, exercise RHC requires operator expertise, is expensive, and has limited availability; the results are also difficult to interpret due to wide swings in intra-thoracic pressures. Moreover, the procedural approach has not been widely standardized above all the hemodynamic measurements and their interpretation. An alternative could be dobutamine infusion during RHC, which may be more practical than exercise stress but needs additional validation studies to define its utility. For these reasons, alternative methods simulating exercise have been proposed mostly in patients who are unable to exercise, i.e., saline loading (fluid challenge test), exercise echocardiography and passive leg raise (PLR). Saline loading can be used instead of exercise, but the quality of the acquired values is inferior to the exercise test [41]. However, it is simpler to perform and easier to reach a PCWP > 18 mmHg immediately after rapid infusion (7 mL/kg), which is considered diagnostic of diastolic LV dysfunction, even if its sensitivity is inferior to the exercise test [51,52].

PLR is a useful method to increase venous return by simulating physical exercise and discovering occult HFpEF [51,52]. Occult HFpEF is determined by a PCWP ≥ 19 mmHg after PLR, with a specificity of 100%, irrespective of diuretic use. While a PCWP < 11 mmHg excludes the diagnosis of occult-HFpEF [53]. Physical stress might be needed if values after PLR are between 11–18 mmHg. Finally, exercise stress echocardiography (the diastolic stress test) can also be considered: an average E/e′ > 15 at peak stress, with or without a peak TR velocity > 3.4 m/s, during the diastolic stress test, is diagnostic of HFpEF [4]. However, the diastolic stress test has limitations. Obokata et al. reported that E/e′ was undetermined in about 20% of patients during peak exercise and that TR velocity was measurable in only 50% of patients [54].

## 4. Right Heart Catheterization in AdHF

### 4.1. The Role of RHC in Advanced HFrEF and Candidacy for LVAD or Heart Replacement Therapy

AdHF is characterized by persistent symptoms and signs of HF despite maximal therapy [1]. The incidence of AdHF is rising, which is likely due to the growing population with HFrEF, the increase in the elderly population, and the advent of new drugs and resynchronization therapy in recent years that have significantly improved HFrEF survival. However, HFrEF prognosis remains poor, with a range of mortality between 25% to 75% [55,56,57]. A hemodynamic status evaluation is essential in this category of patients to better guide the therapeutic strategy. Echocardiography is an essential tool to estimate chamber pressures and cardiac function but has some limitations: right heart-filling pressures are challenging or impossible to measure in patients with mechanical ventilation due to the “curtain effect” of the lung, which reduces the ability of the ultrasound to penetrate to the heart and return to the transducer [58], cardiac devices may generate artifacts, and the patient who underwent cardiac surgery may present habitus that makes impossible the alignment with Doppler waveforms.

RHC has many applications in this setting with patients. Its role has been questioned by The Evaluation Study of Congestive Heart Failure and Pulmonary Artery Catheterization Effectiveness (ESCAPE) trial [2], in which 433 patients with severe symptomatic heart failure despite recommended therapies were enrolled at 26 sites from January 2000 to November 2003. They were randomized to receive therapy guided by clinical assessment plus PAC or clinical assessment alone. The goal in both groups was the resolution of clinical congestion, with additional PAC targets of a PWCP of 15 mmHg and a RAP of 8 mmHg. The primary endpoint was days alive out of the hospital during the first 6 months, while secondary endpoints were exercise, NP, and QoL. The conclusions were that the addition of invasive hemodynamic assessment with RHC to clinical assessment did not improve mortality or reduce hospitalization in patients with advanced HF. 2021 American and European heart failure guidelines clearly define conditions in which RHC is recommended in the context of AdHF. In particular, American guidelines recommend invasive hemodynamic assessment in carefully selected patients with acute HF with persistent symptoms and/or when hemodynamics are uncertain (Class IIa) [59]. European guidelines recommend RHC in patients with severe HF being evaluated for heart transplantation or MCS (Class Ia) [1] (Table 3). In the contest of heart transplant or durable MCS candidacy, the role of the RHC is to evaluate the degree of decompensation and to confirm the indication for heart transplantation or durable MCS [60]. Particularly, CI is the cornerstone parameter to judge the degree of decompensation, guide medical therapy, and determine the heart transplant listing priority in the United Network of Organ Sharing allocation criteria in the United States [61]. Furthermore, systolic pulmonary artery pressure (sPAP) must be accurately analyzed by RHC to evaluate cardiac transplant candidacy. sPAP value can predict the safety of the transplant procedure and its success [62]. Indeed, sPAP > 60 mm Hg, PVR > 5 WU, and TPG > 15–20 mm Hg are considered prohibitive for transplantation [63]. In patients with 50 mmHg < sPAP < 60 mm Hg, 3 WU < PVR < 5 WU, and 15 mmHg < TPG < 20 mmHg, a vasoreactivity test is indicated a vasoreactivity test is indicated. If the baseline PVR of >3 WU decreases after intravenous sodium nitroprusside to <3 WU, the PH is not fixed, and the transplantation is not contraindicated [64] (Figure 4). Additionally, RHC can establish the prognosis of transplant patients. Ghio et al. showed that poor response to vasodilator drugs increased the risk of primary graft failure after a heart transplant [65]. Bellettini et al. studied 657 transplant patients, demonstrating that a low pulmonary artery pulsatility index (PAPi) (<1.68) was associated with primary graft failure or urgent renal replacement therapy [66]. De Groote et al. analyzed 425 consecutive patients who underwent RHC and demonstrated that the combination of PVR (≤ or >3 WU), RAP (< or ≥9 mmHg) and mPAP after the inotropic challenge (≤ or >30 mmHg) was the most powerful predictor of major events [67]. However, resting RHC in chronic HF correlates poorly with exercise capacity [68]. Indeed, between resting parameters, PCWP results is an independent prognostic parameter [69] particularly linked to the pulmonary diffusion capacity in chronic HF patients [70]. In a study of 133 patients undergoing resting RHC and CPET, PCWP has been identified as a predictor of peak maximal oxygen consumption (VO_2_), and an increase of 8 mmHg was associated with a 1 mL/min/kg decrease in peak VO_2_ [71]. Interestingly, in a study that compared chronic HF patients with healthy populations, HF patients increased systemic vascular resistance (SVR) during exercise more than healthy populations with an associated reduced exercise capacity. In general, chronic HF patients increase MAP, stroke volume (SV) and, CO and pulmonary vasodilation less in response to exercise [72]. mPAP changes during exercise significantly correlate with exercise capacity and survival in chronic HF patients. Rieth et al. also [43] demonstrated the usefulness of exercise RHC in chronic HF patients. They found that a change in CO of >1.15 L/min during exercise was associated with better outcomes, with a higher 5-year transplant/VAD-free survival compared to those with a lower change in CO. Pugliese et al. [73] demonstrated that peak cardiac power output (CPO) in addition to resting CPO and peak CPO controlled for left ventricular mass are prognostic parameters in patients with chronic HF, whereas exercise parameters such as CPO correlate stronger with functional capacity [74]. Chronic HF patients underwent exercise using a bicycle ergometer in the supine position. However, alternative methods, like Passive Leg Raise (PLR), are more comfortable in HF or LVAD patients [75]. Patients affected by AdHF could also benefit from LVAD as a bridge to transplantation (BTT), as a long-term therapy in case of cardiac transplant contraindication, and as a bridge to decision (BTD) in case of relative contraindication to transplant and reversible comorbidities [62]. Rose et al., in the REMATCH trial [55], randomized 129 patients ineligible for cardiac transplantation with end-stage heart failure to receive a LVAD or optimal medical management. The patients who received LVAD had a lower risk of death from any cause, with a survival rate that remained high at two years (23% vs. 8% in LVAD and medical therapy, respectively). RHC also has an important role in assessing LVAD candidacy as it estimates some hemodynamic parameters, such as low CO, reduced SvO_2_, and increased PCWP, despite optimized medical therapy. Furthermore, it also distinguishes three different phenotypes: LV failure (with PCWP > 18, PAPi > 1.5), RV failure (with PCWP < 18, PAPi < 1.5), and LV/RV failure (with PCWP > 18, PAPi < 1.5) [76,77,78,79,80,81]. This classification is important because LVAD efficacy depends on adequate preload and so on. RV function and RV failure post-LVAD are associated with higher morbidity and mortality. Therefore, it is important to determine the pre-LVAD characterization of RV function and predictors of RV failure during LVAD support to better select the appropriate patients and to improve outcomes after LVAD [82,83,84,85].

### 4.2. The Role of RHC in Post-Heart Replacement Therapy and LVAD Management

RHC has a key role in the assessment of post-transplant complications. The main complication of the transplant is primary graft dysfunction (PGD), defined as LV and/or RV graft failure occurring in the immediate post-transplant period (within 24 h) in the absence of an anatomic or immunologic etiology [87]; it must be distinct to secondary graft dysfunction where there is a discernible cause such as hyperacute rejection, pulmonary hypertension or known surgical complications. RHC has a key role in its diagnosis and the differential diagnosis between LV and RV graft failure, as they have different treatments [88,89]. LV graft failure might be considered: (1) mild by RAP > 15 mmHg, PCWP > 20 mmHg, CI < 2.0 L/min/m^2^ lasting more than one hour and requiring low dose inotropes; (2) moderate by RAP > 15 mmHg, PCWP > 20 mmHg, CI < 2.0 L/min/m^2^, hypotension with MAP < 70 mmHg and high dose inotropes or newly placed intra-aortic balloon pump (IABP); (3) severe if the patients depend on left or biventricular mechanical support. RV graft failure is defined by RAP > 15 mmHg, PCWP < 15 mmHg, CI < 2.0 L/min/m^2^, TPG < 15 mmHg and/or sPAP < 50 mmHg or need for right ventricular assist device (RVAD) [88]. Furthermore, in patients requiring re-transplantation, it is essential to reassess RVP and reversible pulmonary artery pressures [90,91]. RHC is also recommended 3 to 6 months after LVAD implant to verify PAP normalization and allow the patient to be listed for cardiac transplant [62]. Recent data suggest that routine RHC may not be necessary for asymptomatic patients with normal PVR pre-LVAD implant or PVR < 2.5 six months post-LVAD implantation [62]. The main complication after LVAD implantation is right-HF, occurring in 9% to 42% of cases and increasing morbidity and mortality [92]. Late right-heart failure is frequent in patients with LVAD support and is associated with high mortality and increased incidence of adverse events such as gastrointestinal bleeding and stroke [93]. RV failure with a clinical assessment in post-LVAD patients is difficult, as clinical signs of RV failure such as severe TR, ascites, or increased bilirubin appear very late and increase the mortality after LVAD implantation [94]. In this setting, RHC can determine many hemodynamic parameters that predict post-LVAD early right HF. Preoperative RAP (>15 mm Hg), increased RAP/PCWP ratio (>0.54–0.63), reduced RV stroke work index (RVSWI) (<0.25 mm Hg × L/m^2^) [95], and low CI (<2.2 L/min/m^2^) are historical hemodynamic parameters strongly associated with right failure post-LVAD implantation [92,95]. The PA pulsatility index (PA pulsatility index = [PASP − PADP]/RAP) also decreased and predicted the right HF after LVAD implantation [80,81]. 

Recently, the PAPi, defined as the ratio of pulmonary artery pulse pressure to RAP, has been shown to better predict the risk of RV failure and need for RV assist device support post-LVAD implantation than the RAP/PCWP ratio and RVSWI [80,81]. Indeed, both the RAP/PCWP ratio and RVSWI depend on the RV preload and are influenced by the filling pressures, volume overload, and severe mitral regurgitation (MR), increasing the mean PCWP. PAPi represents a right-sided hemodynamic effectiveness measure without the influence of the left part and predicts RV failure better in patients who are on inotropes pre-LVAD implantation because the inotropes unload the LV, reduce PCWP, increase CI, and thus influence the evaluation of CVP/PCWP and RVSWi [95]. Kochav et al. compared PAPi with traditional indices of RV failure, such as RVSWI and RAP/PCWP, in the ESCAPE trial population [96]. They demonstrated that PAPi was associated with clinical (jugular venous distention, ascites, edema), echocardiographic (right atrial size, vena cava size, and TR velocity), and hemodynamic signs of RV failure (RAP, PCWP). It is also associated with traditional parameters of LV failure, such as LV ejection fraction, PCWP, and CI. Therefore, compared to the RVSWI and RAP/PCWP ratio, PAPi strongly predicts adverse clinical events in patients with advanced HF [96]. 

Muslem et al., employing the INTERMACS definition for Right HF, identified several predictors for RHF, of which the ratio of systolic pulmonary artery pressure to stroke volume (SV) was the strongest hemodynamic predictor [97]. Recently, Rajapreyar et al. [93] analyzed late RHF as a complication after LVAD implantation, which increases morbidity and mortality. They defined late RHF as persistent or new diagnosis of RHF > 90 days after index LVAD implantation, with at least two of the four following characteristics: (1) symptoms or signs of RHF, including elevated jugular venous pressure, hepatomegaly, ascites, and/or peripheral edema, increase or persistent need for high-dose diuretic agents; (2) morphological or functional abnormalities on imaging modalities of the right heart (e.g., echocardiography or electrocardiography-gated contrast-enhanced computed tomography scan): i.e., RA/RV dilation, the left-sided shift of interatrial septum, RV systolic dysfunction, and/or significant TR; (3) invasive hemodynamic signs of RHF (RAP > 16 mm Hg, abnormal RAP/PCWP especially in case of RAP > 10 mm Hg), low PAPI, low cardiac index < 2.2 L/min/m^2^, and/or low venous oxygen saturation (<55%); and (4) biomarkers of right HF: increasing in NP, worsening chronic kidney disease or liver function abnormalities and persistent anemia. There are no standardized echocardiographic parameters to define late right HF after LVAD—indeed, tricuspid annular plane systolic excursion (TAPSE) does not predict RV dysfunction after cardiac surgery due to the change in RV contractile pattern. Hemodynamic parameters can predict the risk of late right HF. In particular, the RAP/PCWP ratio increases in all patients after LVAD, and a RAP/PCWP > 1 suggests significant RHF. A RAP/PCWP > 0.54 increases the risk for post-LVAD right HF. Furthermore, some parameters, such as deep y descents on the RAP waveform, low PAPi, and an increasing gradient between PAP and PCWP with increasing LVAD speed, are associated with RV failure after LVAD [93]. 

RHC, in addition to echocardiography, can diagnose LVAD complications such as pump thrombosis that occur with LV pressure and CI changes [98]. Unlike echocardiography, with RHC, it is possible to recognize suboptimal RAP and PCWP in LVAD patients [99], improving left ventricular filling pressure, which is directly associated with HF hospitalizations [100]. However, if LVAD is used as a BTT, RHC also helps to monitor precapillary PH. RHC has a key role in helping LVAD patients optimize device speed through the evaluation of the CO. However, there is no gold standard for CO assessment after LVAD. 

There is a discrepancy between indirect Fick CO versus thermodilution method CO in LVAD patients with a bias of −0.72 L/min [101]. The thermodilution method overestimates CO with low flow conditions, while the indirect Fick method underestimates CO compared with the flow reported from the outflow cannula. It is likely that continuous rather than pulsatile flow in LVAD patients may also contribute to the error in CO calculations [102]. Furthermore, a hemodynamic ramp test is performed periodically in LVAD patients to evaluate and regulate speed devices. It consists of hemodynamic and echocardiographic parameters. Different protocols exist, and all include LVAD speed at baseline, the LV end-systolic dimension, LV end-diastolic dimension, interventricular septum position, the frequency of atrium-ventricular opening, and the degree of MR and aortic regurgitation and RHC parameters as filling pressures and CO. After these measurements, the LVAD speed is modified (increased or decreased) by 100 rpm and 400 rpm. The test is terminated when the LV end-diastolic dimension is less than 3.0 cm, and LVAD speed is set to CVP less than 12 mm Hg and PCWP less than 18 mm Hg.

Furthermore, the goal is to allow intermittent aortic valve opening and minimal MR. LVAD patients with device speed optimization using a standard ramp test have a lower rate of hospital readmission [100,103]. Finally, in a LVAD patient exercise, RHC has a pivotal role, as increasing the device speed positively correlates with peak VO2 [104]. LVAD implantation has an important effect on exercise paraments, as LVAD patients increase their heart rate (HR), blood pressure (BP), PCWP, CO, RAP, and mPAP during exercise with an incremental increase in the left and right heart-filling pressures with exercise [105]. LVAD patients may increase their CO during exercise by reducing peripheral vascular resistance rather than changes in pump speed [106,107]. However, Brassard et al. [108] demonstrated that CO improvements during light exercise through device speed increases do not appear during strong exercise. Moreover, despite maximal improvements during exercise, filling pressures and mixed oxygen saturations remained high [108]. Thus, it seems that the native heart rate contributes more to CO during exercise than device speed increases.

## 5. Right Heart Catheterization in Acute HF (AHF) and Cardiogenic Shock (CS)

### 5.1. The Diagnostic Role of RHC

AHF is a leading cause of hospitalization and death in patients > 65 years of age [109,110]. AHF is characterized by a gradual or rapid worsening of cardiac function with the onset of signs and/or symptoms of HF leading to hospitalization of the patient or an emergency department visit. AHF may be the acute decomposed HF or the first manifestations of HF. The latter is associated with higher in-hospital mortality [111]. Extrinsic factors may precipitate AHF in patients with pre-existing cardiac dysfunction. Physical examination and non-invasive assessment remain crucial in the diagnostic workup of acute HF. The latest European guidelines for the diagnosis and treatment of acute and chronic HF describe four major types of AHF based on the combination of clinical and instrumental signs [112]: the acutely decompensated HF, the most frequent form of AH, acute pulmonary edema (APE), isolated RV failure, and CS [113]. 

Acutely decompensated HF is present in approximately 50–70% of presentations and typically affects patients with a history of HF. It is characterized by a gradual accumulation of fluids with increased filling pressures (LVEDP and PCWP). Likewise, APE is characterized by increased filling pressures (LVEDP and PCWP) but differs from Acutely Decompensated HF for the timing of onset, within hours, and for the symptoms with a fluid redistribution to the lungs and acute respiratory failure. HF with isolated right ventricle dysfunction is characterized by increased right ventricular (RVEDP) and atrial pressure (RAP), resulting in systemic congestion with increased central venous pressure. RV dysfunction can lead to a reduction in cardiac output due to impaired filling of LV. In acute decompensation without shock, RHC is not routinely recommended based on the pivotal randomized trial ESCAPE, which did not show the potential benefit of routine RHC monitoring in this setting [2,114,115]. However, in selected cases, RHC hemodynamic parameters may play an important clinical role, particularly when diagnosis is uncertain (i.e., restrictive cardiomyopathy vs constrictive pericarditis, coronary artery disease) [86]. However, when AHF patients have a clear diagnosis and an appropriate response to pharmacotherapy, RHC is unnecessary; however, in cases of poor response to therapy or a worsening in the renal or respiratory functions towards CS and the need for early diagnosis and classification of the CS stage, the RHC parameters may be very helpful to optimize the diagnosis and therapeutic management of the patient [1,86].

CS is a syndrome characterized by a high-acuity state of decreased cardiac output resulting in end-organ hypoperfusion and frequently associated with multisystem organ failure and death [116,117,118]. More than half of CS are the result of decompensated HF [119], whereas only 13% of the acute coronary syndrome (ACS) develops in CS [120]. However, the well-known benefits of early revascularization of the culprit lesion in CS complicating ST-elevation myocardial infarction (MI) [121,122], in-hospital and short-term mortality remain elevated up to 50% [123,124,125]. Despite significant heterogeneity based on etiology, the prognosis remains very poor [119]. In this setting, early and correct diagnosis of CS is essential to recognize patients requiring specific therapy. 

In recent years, MCS, such as IABP, Impella^®^, TandemHeart™ and ECMO, are gradually more used in this setting to increase SV and/or to unload LV [126]. In the initial evaluation of CS patients, the first step is to perform a physical examination and electrocardiographic and echocardiographic testing. The diagnosis of CS is made when clinical signs and symptoms of low CO and tissue hypoperfusion are present in the absence of other causes. Patients most commonly present low systolic blood pressure (BP < 90 mmHg), with clinical or laboratory evidence of tissue hypoperfusion (e.g., lactate > 2 mmol/L) [126]. Non-invasive measurements, such as echocardiography measurements, are commonly used to give information regarding hemodynamics in SC patients [127]; however, in this context, a non-invasive assessment may not be accurate and may not recognize dynamic changes typical of these conditions [128]. Specifically, echocardiography has crucial limitations in CS: (a) CS is a dynamic process, but continuous echo is not feasible; (b) ultrasound windows are not always permissible, especially in unstable ventilated patients; (c) diastolic function is unavailable or indeterminate in a substantial number of patients; (d) severe valve disease makes interpretation very difficult; (e) time-consuming calculations; and (f) limited RV assessment. Therefore, invasive hemodynamic assessment using RHC provides a crucial tool for diagnosing and monitoring a patient with CS.

The 2019 Society for Cardiovascular Angiography and Interventions (SCAI) consensus statement [129] on the classification of CS supports the concept that clinical evaluation alone is insufficient to better stratify patients in ascending severity stages. In contrast, invasive evaluation is essential, especially at the more advanced (C–E) stages of CS. Therefore, RHC may recognize different “phenotypes” of CS. Frequently, patients with CS have a “cold and wet” phenotype characterized on RHC as reduced CI, increased SVR, and increased PCWP [130]. However, in an Intensive Care Unit (ICU), it is more frequent to find “wet and warm” patients with reduced CI, elevated PCWP and low-to-normal SVR. These patients with a “mixed shock” due to inflammatory or infective reaction have a worse outcome with a higher risk of mortality. In this setting, RHC has a central role since these patients’ risks can be underdiagnosed with the clinical assessment alone [123,131]. Finally, CS can be characterized by LV, RV, or biventricular dysfunction. LV-dominant CS is characterized by decreased LV function, increased PCWP, and reduced/normal CVP. RV-dominant CS presents with decreased RV function, increased CVP, normal/low PAP, and PCWP with preserved left ventricular function. Biventricular CS persists with reduced biventricular function, reduced MAP, and increased CVP and PCWP. At last, RHC may recognize a pre-shock condition characterized by normotension and initial signs of end-organ hypoperfusion. These patients with late recognition also have the worst outcomes [18,116,132]. Furthermore, RHC is essential to recognize the presence and the degree of RV failure. RV dysfunction is present in up to 40% of ACS and is associated with the worst outcome with high mortality, similar to LV failure [133,134]. Several invasive hemodynamics may differ RV versus LV − CS: RAP > 15 mmHg, RAP/PCWP > 0.8, PAPi < 1, and RVSWI < 600 mmHg × mL/m^2^ are typical of RV failure [77,135]. Among hemodynamic predictors proposed to identify RV dysfunction, a PAPi < 0.9 is a strong predictor of severe RV failure and a worse outcome after MI [79].

### 5.2. The Role of RHC in Guiding Therapy

Invasive hemodynamic assessment can help tailor the therapeutic intervention compared with clinical assessment alone [136]. Data from the ESCAPE trial demonstrated that patients in the interventional group received higher doses of vasodilator therapy with better outcomes [2]. Furthermore, patients treated with invasive hemodynamic assessment received more aggressive and early therapy with temporary MCS, which led to better outcomes [137]. Osman et al. retrospectively analyzed 394,635 CS patients (IHM = 62.565; no IHM = 332.070), demonstrating that invasive hemodynamic monitoring (IHM) was associated with lower in-hospital mortality, higher LVAD and heart transplantation use, without difference between the two groups in terms of vascular complications, major bleeding, and the need for renal replacement therapy. However, patients in the IHM group had a higher percentage of concomitant bloodstream infections, longer length of stay, and higher cost compared to the no-IHM group [138]. Significant evidence shows how invasive hemodynamic data (i.e., CI, CPO, PCWP) are useful both to identify the best therapeutic choice and the need for escalation/de-escalation. Indeed, the use of vasodilators and inotropes leads to an increase in CO and MAP at the expense of an increase in cardiac work with a consequent increase in oxygen demand. Furthermore, an increased oxygen demand can lead to an increase in major arrhythmias and a worsening of CS.

MCS devices increase CO by replacing myocardial work—they behave differently depending on the type. Percutaneous LVAD, such as IABP, leads to a small reduction of LV wall stress and afterload. Support systems such as Impella and TandemHeart reduce LV volume and pressure while increasing in MAP without greatly increasing afterload and, consequently, myocardial oxygen consumption. Finally, venoarterial extracorporeal membrane oxygenation (VA-ECMO) increases pulmonary and systemic circulation by reducing the need for LV and RV contraction, increasing afterload. The use of the RHC can help choose the best among the different types of MCS and the best combination of them if necessary (i.e., VA-ECMO in combination with Impella) [76,132]. RHC hemodynamic parameters can establish the presence of LV failure or associated RV failure to define the most appropriate type of MCS [18]. Indeed, CVP is higher than PCWP in case of RV failure, and a CVP/PCWP > 0.86 suggests impaired RV function. Yet, the PAPi < 0.9 is more specifically associated with RV dysfunction and may suggest the need for RV support. The difference in the type of shock according to the hemodynamic parameters such as cardiac output, PCWP, CVP, and MAP can help physicians choose the device for ventricular support (Figure 5). However, the RHC is useful in choosing the best therapy and monitoring the patient during mechanical and/or pharmacological support therapy. CPO is used to calculate patient risk and as an index of adequate support. PCWP monitoring is useful for diagnosing the type of CS; furthermore, continuous monitoring during MCS indicates the degree of right ventricular decongestion and unloading [18,139].

### 5.3. The Prognostic Role of RHC

Regarding the prognostic role of RHC in CS patients, the ESCAPE trial did not demonstrate that invasive hemodynamic assessment is superior in terms of days alive out of hospital in the first 6 months, but it ruled out CS patients [2]. Conversely, data from the National Cardiogenic Shock Initiative demonstrated in patients with acute myocardial infarction (AMI) complicated by CS that a shock protocol using rapid initiation of MCS and IHM is associated with better outcomes [140]. Furthermore, another shock working group analyzed the trends in RHC use among patients with CS. They retrospectively analyzed patients admitted with CS 2004–2018 and compared the rates use of RHC in three-time intervals: pre-ESCAPE trial (2004–2005), post-ESCAPE trial (2006–2015), and the era of the shock teams’ work (2016–2018), and divided the population in AMI-CS and non-AMI-CS patients. They reported an initial decline in RHC use from 13% in the pre-ESCAPE era to 10% in the post-ESCAPE era, followed by a steady increase in the trends of RHC in CS patients of 17% in the shock teams group. Similar trends were observed in patients with AMI-CS and non-AMI-CS [141]. Invasive hemodynamic parameters also have a prognostic role in patients with CS. Indeed, the SHOCK Trial registry (SHould we emergently revascularize Occluded Coronaries for cardiogenic shocK) [142] enrolled 541 patients with CS whose hemodynamic measurements were made between 6 h before and up to 12 h after shock diagnosis. The cardiac power index (CPI), CO, CI, SV, left ventricular work, left ventricular work index, stroke work, mean arterial pressure, systolic and diastolic blood pressure, coronary perfusion pressure, ejection fraction, and pulmonary artery systolic pressure were associated with higher in-hospital mortality. Among these parameters, CPO, calculated as MAP X CO/451, and CPI are the strongest independent predictors of in-hospital mortality in patients with CS. In particular, CPO < 0.53 W has a high sensitivity and specificity in determining in-hospital mortality [142]. Indeed, cardiac power measurement in patients with acute HF, reflecting cardiac pump function, represents most of the recruitable reserve available during the acute event and may reflect the severity of the patient’s condition. 

Several studies have demonstrated that patients with low body weight have higher mortality after acute MI; thus, CPI may result in a lower association with mortality than CPO [143]. Furthermore, a more recent reanalysis of the ESCAPE study demonstrated that there is a lower 6-month mortality among patients who achieve a PCWP + RAP value below 30 mmHg (*p* < 0.0001) compared with patients who have failed to achieve this target [144]. Several studies suggest that RHC in CS patients is associated with improved outcomes. Garan et al., in a large multicentre registry representing real-world patients with CS, demonstrated the association between RHC use and outcome. They analyzed 1414 patients with CS divided by SCAI stages and by RHC-use group (no RHC data, incomplete RHC data, complete RHC data) prior to initiating MCS. CS patients with complete RHC data were observed to have the lowest in-hospital mortality compared to those who did not. This difference was particularly pronounced in the sickest cohort of patients (SCAI Stages D and E patients). In addition, the authors found that CS patients who did not receive MCS were more likely to have a complete RHC assessment [145]. Rossello et al., in a prospective study, demonstrated that the use of RHC is independently associated with lower short and long-term mortality in patients with CS [146]. More recently, in a retrospective evaluation of the Nationwide Readmissions Database (NRD), Ranka et al. reported the data from 25,840 patients who received RHC (9.6% of the CS population). The RHC patients had a 31% reduction in-hospital mortality and a 17% reduction in 30-day rehospitalization. Furthermore, they received more invasive treatment with temporary MCS during rehospitalization [137]. The RHC group had significantly more comorbidities compared with the non-RHC group, demonstrating improved outcomes mostly in this setting of patients [137]. However, Sionis et al., in the prospective, multicentre European CardShock study of 219 patients with CS, showed that patients in the intervention group (RHC group) had worse baseline profiles and were treated more aggressively with MCS. However, they did not demonstrate the superiority of RHC strategy [147]. Basir et al. used hemodynamic parameters (i.e., CPO) and lactate level as criteria to evaluate escalation/de-escalation of MCS. The authors found that in 171 patients with CS-AMI and early initiation of MCS, CPO and lactate measurements at 12 and 24 h better predicted survival after index procedure. Patients with CPO > 0.6 W and lactate <4 md/dL within 12–24 h of their procedure had a 95% overall survival. These parameters may help clinicians in the escalation of MCS and consideration of LVAD or heart transplant. In patients with CPO <0.6 W and a lactate > 4 md/dL within 12–24 h of their procedure, clinicians may remain aggressive in their care [140]. Furthermore, CPO is superior to BP measurements in CS-AMI. Indeed, in the presence of IABP, it seems that the blood pressure increases without an increase in forward CO [148,149]. Finally, in 13,984 patients with AMI-CS treated with the Impella device, a survival benefit was observed in those who had received invasive monitoring during support [150]. Both American and European guidelines do not recommend routine use of IHM in patients with CS [1,59]; however, several observational studies have demonstrated how RHC improves the outcome in this setting, thus underlying that further randomized controlled trials are needed to investigate the use of the RHC in CS patients.

## 6. Conclusions

HF, in its various clinical scenarios, still poses relevant issues in terms of incidence and prevalence, diagnosis, therapeutical management and prognosis. RHC is the gold standard diagnostic test for PH; however, in its various approaches (at rest, after exercise or stress, in the cath lab or at bedside), it has even a crucial role in the various clinical settings of patients with HF. Indeed, by allowing a comprehensive understanding of the patient’s hemodynamic status, RHC offers precious help for clinicians in diagnosing and managing shock and advanced HF in LVAD or heart transplant candidacy. Furthermore, it is an essential component of the HFpEF diagnostic algorithm. There are, however, safety concerns—it is essential to maintain adequate standards in its use. Training in its execution (catheter insertion, acquisition and interpretation of measurements) should continue to be provided to trainee intensivists/interventionalists and nurses, in patients as well as eventually through simulation, but most of all, the selection of the optimal hemodynamic monitoring technique should be guided by patient condition and by the need for additional measurements in the individual patients.

## Figures and Tables

**Figure 1 diagnostics-14-00136-f001:**
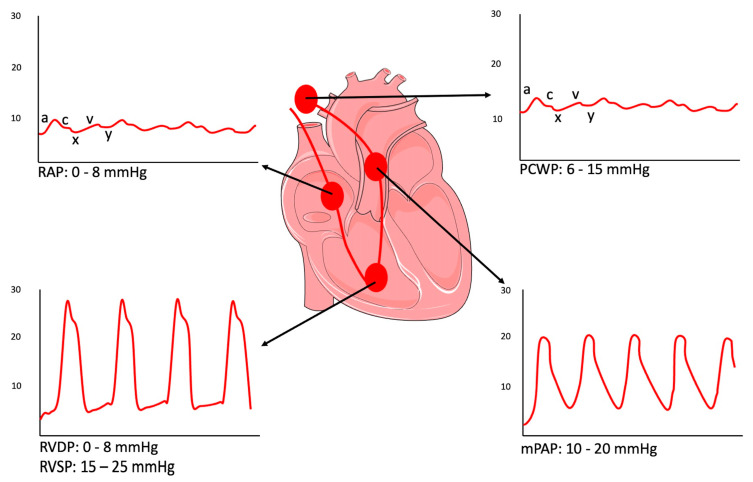
Pressure curves in right heart catheterization. Normal pressure waveforms (a-wave—atrial contraction; c-wave—ventricular contraction; v-wave—atrial filling; x-wave—active atrial emptying; y-wave—passive atrial emptying) obtained in the right atrium (RA), right ventricle (RV), pulmonary artery (PA), and PCWP position with the typical tracings for each; mPAP—mean Pulmonary Artery Pressure; PCWP—Pulmonary Capillary Wedge Pressure; RAP—Right Atrial Pressure; RVDP—Right Ventricular Diastolic Pressure; RVSP—Right Ventricular Systolic Pressure.

**Figure 2 diagnostics-14-00136-f002:**
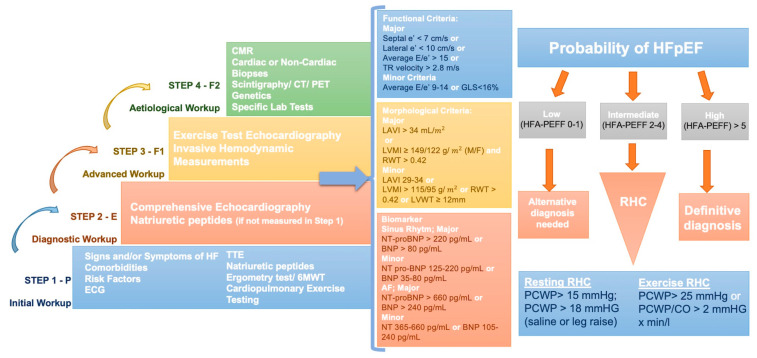
HFA-PEFF stepwise diagnostic algorithm. STEP 1 stands for «Pretest assessment»; STEP 2 stands for «Echocardiography and natriuretic peptides score»; STEP 3 stands for «Functional testing in case of uncertainty»; STEP 4 stands for «Final etiology». AF—Atrial Fibrillation; BNP—Brain Natriuretic Peptide; CMR—Cardiac Magnetic Resonance; CO—Cardiac Output; CT—Computed Tomography; ECG—Electrocardiogram; GLS—Global Longitudinal Strain; HF—Heart Failure; HFpEF—Heart Failure with Preserved Ejection Fraction; LAVI—Left Atrial Volume Index; LVMI—Left Ventricular Mass Index; LVWT—Left Ventricular Wall Thickness; NT-proBNP—N-terminal Pro-Brain Natriuretic Peptide; PCWP—Pulmonary Capillary Wedge Pressure; PET—Positron Emission Tomography; RHC—Right Heart Catheterization; RWT—Relative Wall Thickness; TR—Tricuspid Regurgitation; TTE—Transthoracic Echocardiogram; 6MWT—6-Minute Walking Test. Adapted from Pieske B et al. [4].

**Figure 3 diagnostics-14-00136-f003:**
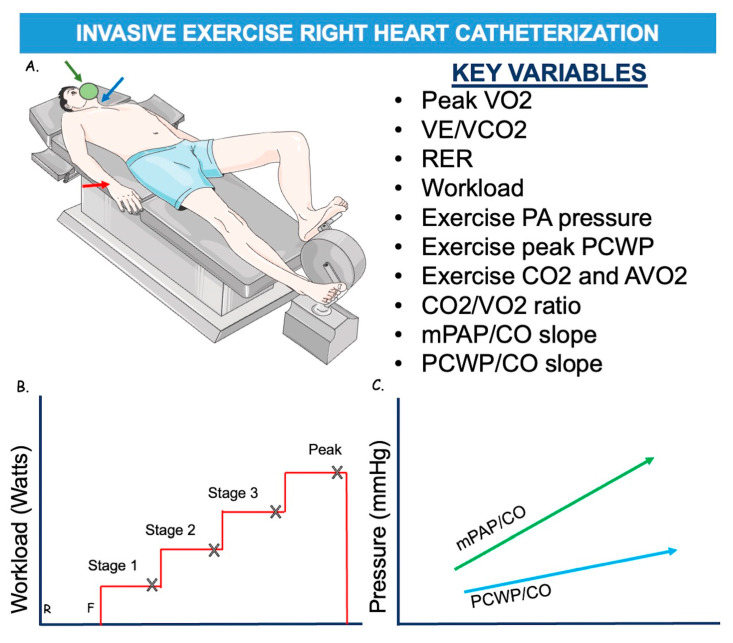
Representation of pressure invasive exercise RHC. (**A**). The patient is usually in a supine position, exercising at a cycle ergometer. It is necessary for artery access, usually radial (red arrow), to assess systemic BP and SaO2, and central venous access, usually through a jugular vein (blue arrow), for the procedure itself. A mask (in green) is used to assess ventilatory parameters. (**B**). Workload scheme. The protocol starts with a cycle unloaded (0 watts) at rest (R) and then starts exercise with feet on a cycle ergometer (F) and increases of 20–25 Watts up to maximal exercise capacity every about 3 min. Measurements are practiced at every stage (X). (**C**). Representation of mPAP/CO and PCWP/CO slopes. AVO_2_—Arteriovenous Oxygen Difference; BP—Blood Pressure; CO—Cardiac Output; CO_2_—Carbon Dioxide; mPAP—Mean Pulmonary Artery Pressure; PA—Pulmonary Artery; PCWP—Pulmonary Capillary Wedge Pressure—SaO_2_—Oxygen Saturation; RER—Respiratory Exchange Rate; VE—Pulmonary Ventilation during exercise; VCO_2_—Carbon Dioxide Production; VO_2_—Venous Oxygen Saturation. Adapted from Hsu S. et al. [10].

**Figure 4 diagnostics-14-00136-f004:**
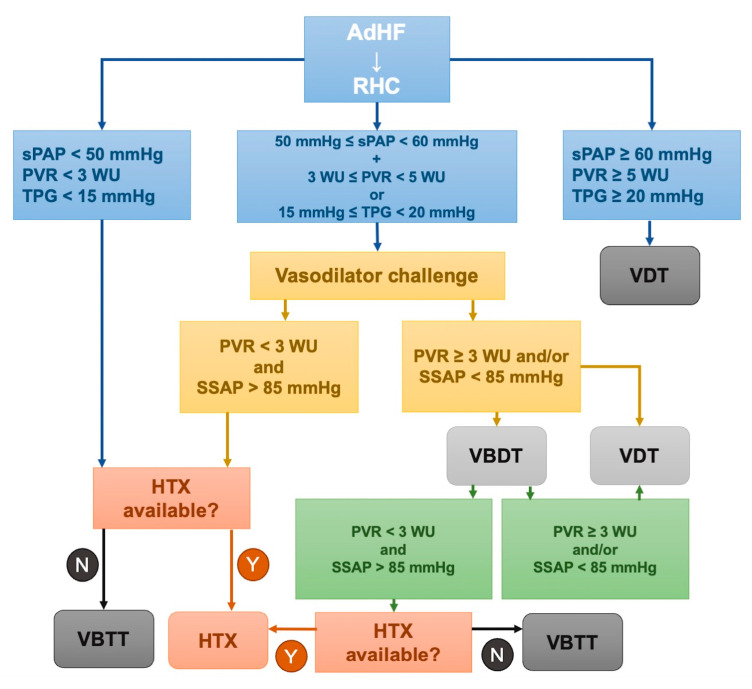
Potential role of RHC to support diagnosis and decision-making in the context of advanced heart failure. Diagnostic algorithm of patients with AdHF candidates for heart transplant. With sPAP < 50 mm Hg, PVR < 3 WU, and transpulmonary gradient < 15 mmHg HTX is possible. sPAP ≥ 60 mm Hg, PVR ≥ 5 WU, and transpulmonary gradient ≥ 20 mm Hg are considered prohibitive for transplantation. In patients with 50 mmHg ≤ sPAP < 60 mm Hg, 3 WU ≤ PVR < 5 WU, and 15 mmHg ≤ transpulmonary gradient < 20 mmHg, a vasoreactivity test is indicated. If baseline PVR of ≥3 WU decreases after intravenous sodium nitroprusside to <3 WU, PH is not fixed, and the transplantation is not contraindicated. AdHF—Advanced Heart Failure; HTX—Heart Transplantation; PVR—Pulmonary Vascular Resistance; RHC—Right Heart Catheterization; sPAP—Systolic Pulmonary Artery Pressure; SSAP—Systemic Systolic Arterial Pressure; TPG—Transpulmonary Gradient; VAD—ventricular assist device; VBDT—VAD in Bridge to Decision Therapy; VBTT—VAD in Bridge to Transplantation Therapy; VDT—VAD in Destination Therapy; WU—Wood units. Adapted from Kittleson et al. [86].

**Figure 5 diagnostics-14-00136-f005:**
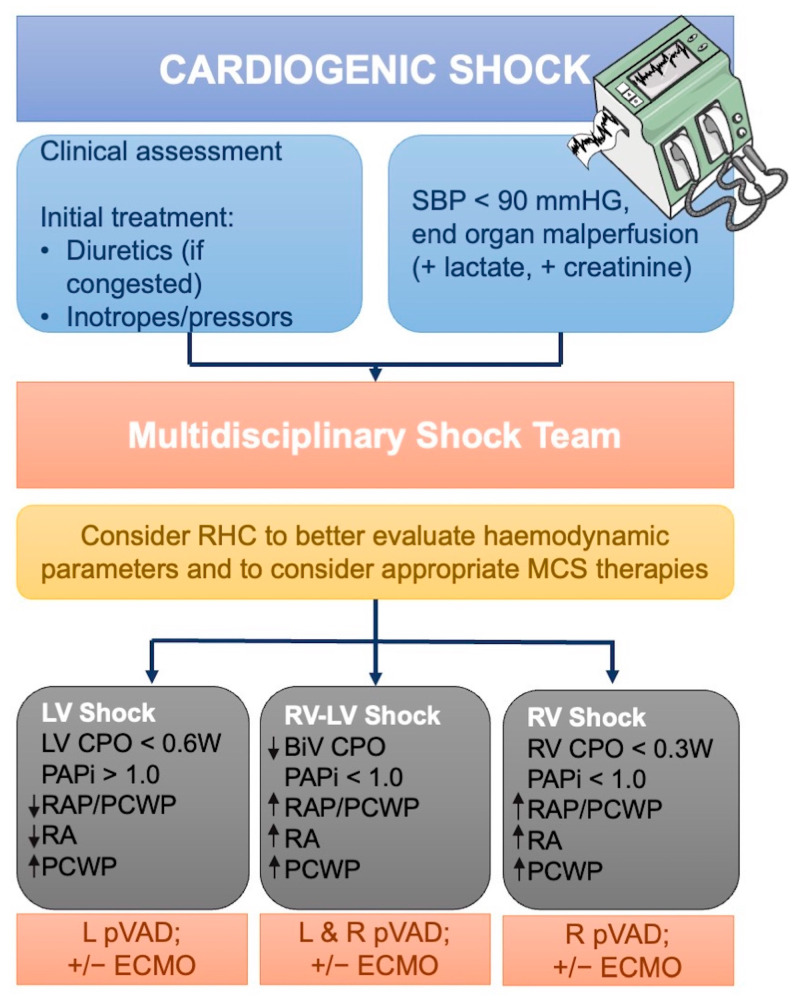
Potential role of RHC to support diagnosis and decision-making in the context of cardiogenic shock. A team-based approach to the management of CS. The difference in the type of shock (LV, RV-LV, or RV shock) according to the hemodynamic parameters such as CPO, PAPi, PCWP, CVP, and mean arterial pressure can help the physicians in the choice of the device for ventricular support. BiV—Bi-Ventricular; ECMO—Extracorporeal Membrane Oxygenation; CPO—Cardiac Power Output; LV—Left Ventricular; MCS—Mechanical Circulatory Support; PAPi—Pulmonary Artery Pulsatility Index; PCWP—Pulmonary Capillary Wedge Pressure; pVAD—Percutaneous Ventricular Assist Device; RA—Right Atrial; RAP—Right Atrial Pressure; RHC—Right Heart Catheterization; RV—Right Ventricular; SBP—Systolic Blood Pressure. Adapted from Hsu S. et al. [10].

**Table 1 diagnostics-14-00136-t001:** Hemodynamic parameters derived from right heart catheterization.

HEMODYNAMIC PARAMETERS	NORMAL RANGE
Central venous pressure (CVP)	2–6 mmHg
Right atrial pressure (RAP)	0–8 mmHg
Right ventricular systolic pressure (RVSP)	15–25 mmHg
Right ventricular diastolic pressure (RVDP)	0–8 mmHg
Pulmonary artery systolic/diastolic pressure (PASP/PADP)	15–25 mmHg; 8–15 mmHg
Mean pulmonary artery pressure (mPAP: PADP + [PASP − PADP/3])	10–20 mmHg
Pulmonary capillary wedge pressure (PCWP)	6–15 mmHg
Diastolic pulmonary gradient (DPG: PADP − PCWP)	<7 mmHg
PAPi ([PASP − PADP]/RAP)	>2
Transpulmonary gradient (TPG: mPAP − PCWP)	<12 mmHg
Mean systemic arterial pressure (mSAP: DBP + 1/3[SBP − DBP])	65–110 mmHg
Cardiac output (CO)	4–8 L/min
Cardiac index (CI: CO/BSA)	2.5–4 L/min/m^2^
Stroke volume (SV)	60–100 mL/beat
Stroke volume index (SVI: SV/BSA)	33–47 mL/m^2^/beat
Systemic vascular resistance (SVR: [mSAP − RAP]/CO × 80)	700–1600 dyn·s/cm^5^
SVR index (SVRI: SVR/CI)	1330–3040 dyn·s/cm^5^·m^2^
Pulmonary vascular resistance (PVR: [mPAP − PCWP]/CO × 80)	20–120 dyn·s/cm^5^
PVR index (PVRI: PVR/CI)	38–228 dyn·s/cm^5^·m^2^

BSA—body surface area; DBP—diastolic blood pressure; SBP—systolic blood pressure. SVR and PVR are reported as dyn·s/cm^5^ and are equivalent to Wood unit by dividing by 80. Normal range of SVR index and PVR index based on an average BSA of 1.9 m^2^.

**Table 2 diagnostics-14-00136-t002:** H2FPEF score description.

H_2_FPEF		
Clinical Variable	Values	Pts
H—Heavy	BMI > 30 Kg/m^2^	2
H—Hypertensive	≥2 antihypertensive drugs	1
F—Atrial Fibrillation	Paroxysmal or Persistent	3
P—Pulmonary hypertension	* sPAP > 35 mmHG	1
E—Elder	Age > 60 years	1
F—Filling pressure	* E/e′ > 9	1

* Assessed by Doppler echocardiography. The H_2_FPEF score includes six clinical variables. A score ≥6 is highly diagnostic of HFpEF. H2FPEF 0–1 score is a lowly diagnostic of HFpEF. For an H2FPEF 2–5 intermediate score, further evaluation is needed. BMI—body mass index; sPAP—Systolic Pulmonary Artery Pressure. Adapted from Kittelson et al. [3].

**Table 3 diagnostics-14-00136-t003:** RHC applications in HFrEF.

Clinical Scenario	Application	Parameters
Acute decompensated HF	- Hemodynamic assessment - Therapy management - Drug titration	CI, CPO, CO, mPAP, PCWP, RAP, sVO_2_, TPG
Heart Transplant eligibility	- Transplant benefits assessment - Surgery safety estimation - Peri- and post-operative mortality estimation - Vasoreactivity test	CI, CO, mPAP, PAC, PAPi, sPAP, PVR, RAP, TPG
Post-Heart Transplant Evaluation	- Heart transplant response assessment - PGD evaluation - Need for myocardial biopsy assessment - Need for retransplantation evaluation	CI, sPAP, PCWP, PVR, RAP, TPG
VAD candidacy	- Need for VAD assessment (INTERMACS profile) - Device to be implanted choice (LVAD/RVAD/BiVAD) - Post LVAD implant RHF probability estimation	CI, CPO, CO, mPAP, PAPi, PCWP, RAP, RVSWI, sVO_2_, TPG
Post-VAD Evaluation	- VAD response assessment - Heart transplant revaluation - Need for device upgrade/shift (shift from LVAD/RVAD to BiVAD, shift from LVAD to RVAD or vice versa) evaluation	CI, CPO, CO, mPAP, PAC, PAPi, PCWP, PVR, RAP, sVO_2_, TPG

BiVAD—Bi-Ventricular Assist Device; CI—Cardiac Index; CO—Cardiac Output; CPO—Cardiac Power Output; HF—Heart Failure; LVAD—Left Ventricular Assist Device; mPAP—Mean Pulmonary Artery Pressure; PAC—Pulmonary Artery Compliance; PAPi—Pulmonary Artery Pulsatility Index; PCWP—Pulmonary Capillary Wedge Pressure; PGD—Primary Graft Disfunction; PVR—Pulmonary Vascular Resistance; RAP—Right atrial pressure; RHF—Right Heart Failure; RVAD—Right Ventricular Assist Device; RVSWI—Right Ventricle Stroke Work Index; sPAP—Systolic Pulmonary Artery Pressure; sVO_2_—Mixed Venous Oxygen Saturation; TPG—Transpulmonary Gradient; VAD—Ventricular Assist Device.

## Data Availability

Not applicable.

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
