# Peer review of "Contemporary Evidence and Practice on Right Heart Catheterization in Patients with Acute or Chronic Heart Failure"

_diagnostics, 2024, doi:10.3390/diagnostics14020136_

Round 1
Reviewer 1 Report
Comments and Suggestions for Authors
The aim of this Paper is to review the evidence on Right heart catheterization in various clinical scenarios of patients with Heart Failure.
The paper is a bit confusing and above all the objectives are not clear.
The introduction is too short, and the purpose of the review is unclear.
Why do the authors do this review? What do you want to focus on?
Even in the conclusions this is not evident.
Figure 2 is a table.
It is not always clear whether the figures are taken from cited articles.
Why are the conclusions after the figures?
Reviewer 2 Report
Comments and Suggestions for Authors
Thank you for the opportunity to review this outstanding paper, which offers contemporary insights into right heart catheterization in patients with acute or chronic heart failure. I have a few minor comments and suggestions for your consideration:
- In several instances, the decimal points are represented by commas. For example, “<0,01%” on line #258. Could you please replace these commas with full stops for consistency?
- I recommend changing the font from 'Comic Sans' to 'Arial' in the tables and figures. Arial tends to be more professional and is easier to read in academic contexts.
- Regarding the algorithms presented in Figures 3, 4, and 6, could you confirm if these are original concepts developed by the authors? If they are not, providing the appropriate references for these algorithms would be beneficial.
Reviewer 3 Report
Comments and Suggestions for Authors
A good quality work, well structured and explained, with quality imaging and impactful information for the clinician. Congratulations to the authors.
Author Response
We thank the Reviewer for the time and efforts invested in the revision of our manuscript and for the positive comments provided.
